# Synchronous Changes of GPP and Solar-Induced Chlorophyll Fluorescence in a Subtropical Evergreen Coniferous Forest

**DOI:** 10.3390/plants12112224

**Published:** 2023-06-05

**Authors:** Mingming Wang, Leiming Zhang

**Affiliations:** 1Key Laboratory of Ecosystem Network Observation and Modeling, Institute of Geographic Sciences and Natural Resources Research, Chinese Academy of Sciences, Beijing 100101, China; 2College of Resources and Environment, University of Chinese Academy of Sciences, Beijing 100190, China

**Keywords:** SIF, GPP, dynamic changes, environmental responses, temporal scale

## Abstract

Using in situ near-surface observations of solar-induced chlorophyll fluorescence (SIF) and gross primary productivity (GPP) of a subtropical evergreen coniferous forest in southern China, this study analyzed the dynamics of SIF, GPP and their environmental responses, and explored the potential of SIF in characterizing the variation of GPP. The results showed that SIF and GPP have similar diurnal and seasonal variation and both reach the highest value in summer, indicating that the SIF can be applied to indicate the seasonal variation of GPP for the subtropical evergreen co-niferous. With the increase in temporal scale, the correlation between SIF and GPP becomes more linear. The diurnal variations of both SIF and GPP were characterized by photosynthetically active radiation (PAR), the seasonal variations of SIF and GPP were influenced by air temperature (Ta) and PAR. Probably due to the absent of drought stress during the study period, no significant correlation was detected between soil water content (SWC) and either SIF or GPP. With the in-crease in Ta, PAR or SWC, the linear correlation between the SIF and GPP gradually decreased, and when Ta or PAR was relatively higher, the correlation between SIF and GPP become weakly. Further research is still needed to illustrate the relationship between SIF and GPP under drought condition which occurred frequently in this region based on longer observation.

## 1. Introduction

Gross primary production (GPP) can reflect the total organic carbon fixed by vegetation through photosynthesis, directly driving the ecosystem carbon cycle and directly or indirectly affecting the function, biodiversity, and sustainability of terrestrial ecosystems [1,2]. Therefore, the accurate understanding and rapid assessment of GPP and its dynamic changes are indispensable to understand the response and adaptation of ecosystem to environmental changes, and to provide important support for research of global change and ecosystem service across different time and space scales [3]. Solar-induced fluorescence (SIF) is assumed widely related to actual plant photochemistry. When chlorophyll absorbs sunlight, it rises from the stable and lowest energy state to an unstable and high-energy excited state, and then quickly turns to the low-energy state, and the emitted energy is consumed in the form of heat and fluorescence [4]. Under natural light conditions, chlorophyll fluorescence accounts only for 0.5–2% of sunlight energy absorbed by vegetation, and the amount of chlorophyll fluorescence emitted is directly proportional to that of photosynthesis. Therefore, variations related to plant photosynthesis can be potentially detected by fluorescence [5], and SIF is also considered to be an effective indicator of plant photosynthesis [6].

In recent years, the estimation of GPP based on observed SIF has received more attention. Some studies found that the SIF obtained via GOSAT satellite showed a strong linear correlation with GPP [7,8]. Li et al. carried out the first global analysis of the relationship between SIF from OCO-2 satellite and GPP from flux observations, and they found that SIF and GPP showed a strong linear correlation among eight ecosystems, in which the slope of evergreen broad-leaved forests was the lowest, and the slopes of evergreen coniferous forests and deciduous broad-leaved forests were similar [9]. Using SIF from GOME-2 satellite, Liu et al. indicated that satellite SIF were significantly correlated with GPP of different ecosystems in China, and the slopes were the lowest in both temperate coniferous and broadleaved mixed forests, while higher in evergreen forests [10]. However, the satellite SIF was utilized in most studies to examine the relationship between SIF and GPP, the utilization of local observed SIF was still deficient. 

The relationship between SIF and GPP differs greatly in different vegetation types mainly due to the influence of canopy structure, vegetation physiological status, and the environmental variations. Yang et al. found that the seasonal variation of SIF and GPP were similar which yielded a linear correlation in a temperate deciduous forest [11]. Magney et al. found that SIF can be used to characterize the variation of GPP, while the slope is relatively low in subalpine coniferous forests [6]. Jeong et al. analyzed the phenology of high-latitude forest vegetation in Eurasia and North America, and they found that the spring SIF was lagged behind that of the vegetation index due to the limitation of the available radiation on carbon assimilation, while a better consistency between GPP and SIF was exhibited in autumn [12]. Cheng et al. found a weak correlation between daily SIF and GPP, and such a correlation was not significant in diurnal scale and in sunny conditions [13]. In recent studies, SIF has been widely used to estimate regional and global GPP, but some studies have shown that GPP will be saturated under strong light, resulted in a nonlinear relationship between GPP and SIF in short time scales, which becomes more linear with the increase in spatial and temporal scales [14,15,16]. At present, most studies were limited to middle and high latitudes, the observations in low latitude ecosystems, especially the coniferous forest, remain relatively lack. Therefore, the capability of SIF to characterize the variations of GPP change needs further studies across more ecosystems and regions.

The plantation area in China ranks among the top in the world, and plantation plays an important role in terrestrial carbon sink capacity of China [17]. The evergreen coniferous plantation is widely distributed in China’s subtropical region. Recent study has indicated that these plantations played an important role in the formation of carbon sink in East Asian under the monsoon climate, and the annual net carbon sequestration reached 387.2 gC m^−2^ a^−1^ [18]. Apart from the abundant rainfall and heat resources in this region, it suffered high temperature and seasonal drought in summer frequently, which resulted in a significant effect on the ecosystem productivity [19,20]. Chen et al. also found that the rising temperature would increase the NPQ of the leaves and lead to a decline in SIF and GPP [18]. Therefore, the accurate understanding and rapid detection of the changes in plantation productivity are highly significant for the assessment of regional carbon sink.

Using the collaborative observation of canopy SIF vegetation, ecosystem CO_2_ flux and environmental factors, the dynamic changes of both SIF and GPP of a subtropical coniferous plantation were analyzed. The targets of this study aimed to (1) reveal the seasonal variations of both GPP and SIF; (2) comparison the responses of SIF and GPP to environmental changes, and (3) examine the potential of SIF in charactering the variation of GPP.

## 2. Results

### 2.1. Dynamic Changes of SIF and GPP

#### 2.1.1. Diurnal Variations

Figure 1 shows the diurnal variations of SIF, GPP and local environmental variables in different seasons during the study period of the subtropical evergreen coniferous forest in Qianyanzhou. It can be seen that in each season, the diurnal variation of GPP showed a single peak trend which reached the maximum value at 13:00 in winter, 12:30 in spring and 10:00 in summer. In addition, the time span of active GPP in a day gradually increased from winter to summer. The diurnal variation of SIF in each season was similar to that of GPP. The maximum value was close in spring and summer, and both were significantly higher than that in winter. The maximum value of SIF appeared at 10:00 in winter, 12:30 in spring and 12:30 in summer.

For the diurnal variation of the environmental factors, Ta also showed a single peak trend, while the time when Ta reached its maximum value apparently lagged behind that of SIF and GDP in each season. PAR also appeared a similar single peak trend with that of GPP and SIF. In terms of SWC, although an apparent variation was presented in each season due to the data standardization, the measured SWC was quite stable in a day and the average SWC in summer was 0.35 m^3^ m^−3^ which was slightly lower than that in spring (0.38 m^3^ m^−3^) and fall (0.39 m^3^ m^−3^).

From the general diurnal variation from January to August, both SIF and GPP showed a trend of increasing first and then decreasing (Figure 1d). The vegetation began to produce SIF from the morning, and the maximum value was reached at about 12:30; then it began to decline, and SIF stopped being produced under low PAR in late afternoon. GPP began to increase at about 5:00 a.m. and reached the maximum value at about 12:30; then it began to decline and dropped to 0 when PAR was quite low. It can be seen that the daily maximum values of SIF and GPP were close to the time of decline. 

Ta also showed a single peak trend, reaching the minimum value at 6:30 and the maximum value at 16:00, lagging behind the time when SIF and GPP reached their maximum values. PAR showed a similar trend with GPP and SIF, reaching the maximum value at 13:00, which was close to the time when GPP and SIF reached their maximum values.

In general, the similar diurnal variation of SIF and the GPP indicated the potential of SIF to effectively reflect the variation in GPP. The diurnal variation in Ta also presented a single peak trend, but the time when it reached the maximum value obviously lagged behind SIF and GPP. PAR, GPP and SIF showed similar change trends, and the times when they reached the maximum values were similar. Therefore, the diurnal variations of both SIF and GPP were determined by PAR. Because SWC displayed no obvious diurnal variation characteristics, implying that SWC probably has a small impact on diurnal variation of both SIF and GPP at Qianyanzhou Station.

#### 2.1.2. Seasonal Variations

Figure 2 shows the seasonal variations of SIF, GPP and environmental variables at Qianyanzhou Station. It can be seen that the SIF value showed a gradual upward trend from spring to summer and tended to be stable in June. It reached the minimum value of 0.08 W m^−2^ μm^−1^ sr^−1^ at the end of February and attained the maximum value of 0.46 W m^−2^ μm^−1^ sr^−1^ by the end of June. GPP showed the same trend as that of SIF. It also increased gradually from spring to summer and tended to be stable in June. The GPP value was the minimum value of 2.28 gC m^−2^ d^−1^ at the end of February and the maximum value of 8.16 gC m^−2^ d^−1^ at the beginning of June. Therefore, the consistency between seasonal SIF and GPP indicated SIF can effectively characterize the seasonal change in GPP in the subtropical coniferous ecosystem.

For the seasonal trends of environmental variables, including Ta, PAR, SWC and rainfall, it can be seen that Ta gradually increased from spring to summer and tended to stabilize in June. Ta reached the minimum value of 11.07 °C at the end of January and the maximum value of 31.54 °C at the beginning of July. We found that PAR showed a relatively stable trend from January to April, increased slightly in May, and tended to be stable in June. PAR reached the minimum value of 77.00 μmol m^−2^ s^−1^ at the end of February, reaching the maximum value of 433.75 μmol m^−2^ s^−1^ at the beginning of July. SWC was mainly affected by precipitation and generally showed a slight increase after the beginning of April and a gradual decrease after the beginning of June. SWC reached the maximum value of 0.40 m^3^ m^−3^ at the end of May, and the value gradually decreased from June to the minimum value of 0.22 m^3^ m^−3^ at the beginning of September.

In general, the seasonal trend of Ta was similar to that of GPP, which both increased from spring to summer and tended to be stable in June, while the increasing trend of Ta was relatively more obvious. PAR showed a relatively stable change from January to April, rose in May, and tended to be stable in June, which was slightly different from the change trend of GPP. SWC was relatively stable from January to March and began to rise in April, reached its maximum value in May, and gradually declined after June, which was apparently different from the change trend of GPP.

### 2.2. Correlation between SIF and GPP

In the subtropical evergreen coniferous forest, the correlation between the SIF of vegetation and GPP became more significant with the increase in temporal scales (Figure 3). On the daily scale, SIF and GPP showed a nonlinear correlation, and SIF only explained 23% of the changes to GPP. On the 8-day scale and the monthly scale, the correlation between SIF and GPP became more linear. On the 8-day scale, the correlation between SIF and GPP was 0.60, and the slope was 12.50. On the monthly scale, the correlation between SIF and GPP attained 0.76, and the slope was 18.43.

### 2.3. Environmental Responses of SIF and GPP

Figure 4 shows the correlation between daily SIF, GPP and environmental variables. The regression analysis showed that the influence of Ta on SIF reached 20%, while the influence of PAR on SIF was 15%, which was slightly lower than that of Ta. Both Ta and PAR had strong correlations with GPP. Ta had an impact on GPP of 67%, while the influence of PAR on GPP increased to 79%. However, GPP appeared to be saturated when PAR was much higher. In addition, SWC had no significant impact on either SIF and GPP.

In order to explore the impact of different environmental changes on the relationship between SIF and GPP, in this study, we obtained the changes in the relationship between SIF and GPP under different environmental which were divided into groups with 30 data. Figure 5 shows that when Ta was low, SIF had a strong relationship with GPP. When Ta was between 8.8 and 13.6 °C, the performance was the strongest. After that, the relationship between SIF on GPP decreased. When Ta attained 22.1–28.2 °C and 29.7–32.5 °C, no significant linear correlation was detected.

The impact of PAR on the correlation between SIF and GPP also showed a trend of decreasing linear correlation between SIF and GPP with the increase of PAR. When PAR was at 21.29–79.78 μmol m^−2^ s^−1^, SIF appeared the strongest indication of GPP. When PAR was greater than 362.84 μmol m^−2^ s^−1^, there was no significant linear correlation between SIF and GPP.

When SWC was 0.20–0.30 m^3^ m^−3^, SIF had no significant linear correlation with GPP. When SWC was 0.30–0.35 m^3^ m^−3^, SIF had the strongest indication effect on GPP, and then, with the increase in SWC, such effect decreased dramatically. This reduction may have been related to the increase in precipitation during this period, which led to the lower radiation and affected GPP.

It can be seen that Ta and PAR had a significant impact on the changes in SIF and GPP, while SWC did not have a significant impact on either. Regarding the correlation between SIF and GPP, except when SWC was 0.20–0.30 m^3^ m^−3^, SIF had no significant linear correlation with GPP, and the linear correlation between SIF and GPP gradually decreased with the increase in Ta, PAR or SWC. When Ta or PAR was much higher, no significant linear correlation appeared between SIF and GPP.

## 3. Discussion

### 3.1. Dynamic Changes in SIF and GPP

Previous studies have revealed the dynamic characteristics of SIF and GPP of canopy at different temporal scales, which showed that SIF and GPP of a coniferous forest ecosystem had obvious diurnal and seasonal changes, and both were the highest in summer and the lowest in winter [11]. In this study, we also found that the SIF and GPP a subtropical coniferous plantation appeared the similar diurnal and seasonal variation, and both reached their highest in summer.

The strongest solar radiation in a day occurs at around 12:00, but the atmosphere is not directly heated from the solar radiation, because the solar radiation warms the ground first, absorbs the heat, reflects back to the atmosphere, and then warms up; therefore, the strongest solar radiation appeared at noon, and the highest atmospheric temperature occurs in the afternoon. In this study, we found that the times when PAR, SIF and GPP reach their maximum values are similar, while Ta lags behind these changes. This shows that SIF and GPP mainly respond to changes in PAR, which is the same conclusion as those drawn in previous studies [18].

### 3.2. Relationship between SIF and GPP

Under natural light, the chlorophyll fluorescence emitted by vegetation is significantly correlated with the photosynthetic rate; therefore, SIF is assumed to has the potential to characterize GPP [5,6]. Previous studies showed that there is a significant correlation between SIF and GPP at different temporal and spatial scales [8,9,21,22], but with the decrease in temporal scales, the linear correlation between SIF and GPP gradually weakens [14,15,16]. 

The nonlinear relationship between daily SIF and GPP can be probably explained in the following reasons. First, there is the basic difference between the light response of SIF and GPP [14]. Photosynthetic effective radiation absorbed by leaves can be used to drive photochemical reactions, which represent the key driving factor of SIF and GPP [4]. When the light intensity is too high and the light energy received by a plant’s leaves exceeds its utilization capacity, photoinhibition may occur, making GPP saturated or even decreased. While SIF will maintain an increasing trend, meaning that SIF-GPP presents a nonlinear relationship on the daily scale [14,15,16]. In this study, we also found that the linear correlation between SIF-GPP gradually decreased until it was not significant when PAR increased continuously. With the increase in temporal scale, the saturation effect of GPP with the increase in PAR decreases (Figure 6), which results in a more linear correlation between SIF and GPP [16,22,23].

Second, the vegetation canopy structure has a potential impact on the SIF-GPP relationship. The positive correlation between the fluorescence yield (SIF_yield_) and the light energy utilization efficiency (LUE) affected by the vegetation canopy structure has been verified in different ecosystems, partially explaining the internal reason for the linear relationship between SIF and GPP [24,25,26].

At last, tower-based spectrometers mainly receive SIF signals emitted from the top canopy leaves and from the internal canopy, which is different from the GPP from flux observation. As a result, the relationship between SIF and GPP might be influenced by the observation approaches, and results in the low correlation at shorter temporal scales. As the temporal scale increases, this influence becomes weaker, thereby the linear correlation between the GPP and SIF is also improved [27].

### 3.3. Impact of Environmental Factors on the Relationship between SIF and GPP

Previous studies showed that the relationship between SIF and GPP is affected by environmental factors [28,29]. Some studies have indicated that under drought or high-temperature conditions, vegetation photosynthesis is affected by climate, and SIF shows a significant decline, reflecting the high sensitivity of SIF to climate change. Therefore, climate change also affects the correlation between SIF and GPP [30,31].

The light energy absorbed by plants is mainly used for three parts: photochemical reactions, heat dissipation, and chlorophyll fluorescence [4]. Research has shown that vegetation SIF mainly responds to changes in PAR, rather than actual changes in photosynthetic efficiency, and the key driving factor for SIF and GPP is photosynthetic effective radiation [27]. This study also shows that PAR has a significant impact on both SIF and GPP. In addition, the radiant intensity of the sun will also cause the change of ground temperature, so Ta also has a significant impact on SIF and GPP.

SWC is usually treated as an important indicator that reflects the variations of GPP when the vegetation is under drought stress. Previous research shows that plant photosynthesis efficiency decreases due to the reduction of transpiration rate of leaves when drought occurs [32,33]. However, the influence of SWC on SIF or GPP was not detected in this study. Probably, under the subtropical monsoon climate with abundant precipitation during the study period, no apparent drought occurred and the relative sufficient soil moisture content, the photosynthetic rate and chlorophyll fluorescence production of vegetation were not significantly affected by the variations of SWC. At the same time, there are still many deficiencies in this study, such as how SIF characterizes the changes in GPP under extreme drought and high temperature stress which occurs frequently. Further research is needed based on longer observation in the future.

There are also some factors that can affect the SIF-GPP relationship, such as vegetation physiological status, chlorophyll content, canopy structure, water and dry matter [11,24,34,35]. When the time resolution of data is reduced to a low resolution, i.e., the increase in temporal scale, the influence of these factors is reduced to some extent, and the linear correlation could be enhanced.

## 4. Materials and Methods

### 4.1. Study Area

Qianyanzhou Ecological Research Station is located in Taihe County, Jiangxi Province (26°44′29″ N, 115°3′29″ E), which is located in the southern hilly area in China. It is a typical red soil area with subtropical monsoon climate characteristics. The region is rich in solar energy resources, with annual average sunshine of 1306 h. The annual average temperature is 17.9 °C. The annual average rainfall can reach 1490.5 mm. Apart from the abundant rainfall and heat resources in this region, due to the uneven seasonal distribution and the incomplete synchronization between rainfall and temperature, the influences of high temperature and seasonal drought in summer occurred frequently [20]. The vegetation in this area is evergreen coniferous plantation in the middle subtropical zone in China, which is mainly composed of secondary vegetation of artificial forest or grass and shrub, including *Pinus massoniana, Pinus elliottii, Cunninghamia lanceolata*, etc. [19]. The forest in this area was planted in the 1980s, with a forest age of approximately 40 years (Figure 7).

### 4.2. Flux Observation

This study directly utilized the shared data of ChinaFLUX (http://www.chinaflux.org/ (accessed on 1 March 2023)). Eddy covariance technique (EC) is generally utilized to measure the Net Ecosystem Exchange (NEE) between vegetation and the atmosphere [36]. In this study, the open-path Eddy covariance (OPEC) is adopted for flux observation. The system is composed of a three-dimensional ultrasonic anemometer (CAST3, Campbell Scientific Ltd., Logan, UT, USA) and a fast-response infrared CO_2_/H_2_O analyzer (Li-7500A, LiCor Inc., Lincoln, NE, USA). The data collector (CR5000, Campbell Scientific Ltd., Logan, UT, USA) collects data at a frequency of 10 Hz and the half-hourly average flux was computed and stored in real time. The net ecosystem carbon flux is calculated as:(1)$$F_{c}=\bar{w^{\prime }{\rho_{c}}^{\prime }}$$
where *F_c_* is CO_2_ flux derived from EC; *w*′ is the instantaneous vertical wind velocity fluctuation, and *ρ_c_*′ is the CO_2_ density fluctuation in the air. The upper horizontal line represents the covariance between the instantaneous fluctuation of vertical wind speed and the CO_2_ density during a certain period (30 min).

The standard procedure developed by ChinaFLUX was applied to the measured flux to carry out a series of correction and quality control and generate the half-hourly averaged flux data including coordinate rotation, WPL correction, frequency response correction and night data correction [37]. Consider the effect of the canopy, the storage flux was also estimated from the variation of CO_2_ concentration, and net ecosystem exchange (NEE) was the sum of the measured *F_c_* and the storage flux. After that, the spurious half-hourly CO_2_ flux data were also detected and eliminated using both a flux threshold (|NEE| > 3.0 mg CO_2_ m^−2^ s^−1^) and the algorithms proposed by Papale et al. [38]. The data under stable conditions was also excluded with a reasonable u^∗^ threshold which was helpful to reduce the effect of insufficient fetch and turbulence.

For missing flux and meteorological observation data in a short period of time (less than 2 h), linear interpolation is used; For meteorological data that has been missing for a long time, use meteorological station observation data (excluding SWC and Rain) for interpolation; If the interpolation cannot be completed, use the average daily variation method to complete the data interpolation. For CO_2_ flux data missing for a long time, non-linear regression method is used for interpolation, the Lloyd and Taylor equation is used for nighttime data interpolation, and rectangular hyperbola equation is used to fill the daytime missing data [39].

The marginal distribution sampling method was applied for flux partitioning. Firstly, using the same regression equation as when imputing missing data, the coefficients in the ecosystem respiration equation are determined for nighttime observation data; Then, use this equation to calculate the ecosystem respiration (Reco) at night and during the day; Finally, complete and continuous ecosystem GPP observation data were obtained using daytime CO_2_ flux data and calculated ecosystem respiration [39]. Consider the sign of NEE, negative NEE which indicating carbon absorption by ecosystem, GPP was calculated as,
GPP = R_eco_ − NEE(2)

Further details for flux processing in this station were described in Han et al. [40].

### 4.3. SIF Observation

Using the flux tower of the subtropical evergreen coniferous forest in Qianyanzhou station, an automatic observation system for the solar-induced chlorophyll fluorescence of vegetation was installed at a distance of 30 m from the canopy. The front ends of two optical fibers were installed on the optical fiber fixing device to ensure that the top and bottom were as vertical as possible. The upward optical fiber was connected to a cosine corrector (CC-3) to accurately measure the solar incident spectrum, and the other downward optical fiber was used to observe the vegetation canopy reflection spectrum, and the approximate area of the ground measured by the downward optical fiber of the equipment is 100 m^2^. The spectrometer recorded two downward solar illumination spectral signals and one upward ground reflection spectral signal in the order of solar incidence, ground reflection and solar incidence. The equipment installation is shown in Figure 8.

The vegetation SIF observation data from 27 January to 15 September 2021, including the solar incident radiation at 730 nm to 782 nm and the vegetation-canopy-reflected radiation were obtained. In order to reduce the large deviation in downward radiation and fluorescence due to the limitation of the view angle when the solar altitude angle was low, SIF observation started at 9:00 and then continued to record until 17:30, and the time interval was 7 min in each day. As the 3FLD (3-Fraunhofer Line Depth) algorithm effectively reduces the error caused by the assumption that the fluorescence and reflectivity remain constant in the spectral absorption band, and its operation is relatively simple [41], 3FLD method was also applied in this study to determine SIF. After that, the daily, 8 days and monthly average SIF were calculated for further analysis.

### 4.4. Environmental Observation

Routine meteorological factors, including photosynthetically active radiation (Model LI-190SB, Li-Cor Inc., Lincoln, NE, USA), air temperature (Model HMP45C, Vaisala, Helsinki, Finland), precipitation (Model 52203, Young Co., Traverse City, MI, USA), and soil water content (Model CS616, Campbell Scientific, Logan, UT, USA). The depth of measuring surface soil temperature was 5 cm. The daily, 8 days and monthly average SIF were also calculated. Due to the malfunction of sensor, soil water content during January to March was abnormal and was excluded in the analysis.

### 4.5. Data Processing

In this study, January to February was defined as winter, March to May was defined as spring, and June to August was defined as summer. In order to present the diurnal variations of different variables, we normalized data according to Equation (3) to analyze the diurnal variation of the variables in different seasons.
(3)$$X=\frac{X_{i}-X_{\min}}{X_{\max}-X_{\min}}$$
where *X* is the normalized value, *X*_i_ is the original value and *X*_max_ and *X*_min_ are the maximum and minimum values of the data, respectively.

## 5. Conclusions

Based on the in-situ observation of canopy SIF, GPP and meteorological factors of a subtropical evergreen coniferous forest at Qianyanzhou Station, the relationship between the SIF and GPP, and their responses to environmental changes were analyzed. The following conclusions were drawn:

(1) Both SIF and GPP of the subtropical evergreen coniferous forest presented similar diurnal and seasonal variations, and both reach their highest values in summer, indicating that the potential of SIF in characterizing the variations of GPP.

(2) Along with the increase in time scales, the correlation between the SIF and GPP became more linear.

(3) Ta and PAR significantly affect the changes in the SIF and GPP at different time scales, while SWC has no significant impact on them. With the increase in Ta, PAR or SWC, the linear correlation between SIF and GPP decreases gradually. When Ta or PAR is high, there is no significant linear correlation between SIF and GPP.

## Figures and Tables

**Figure 1 plants-12-02224-f001:**
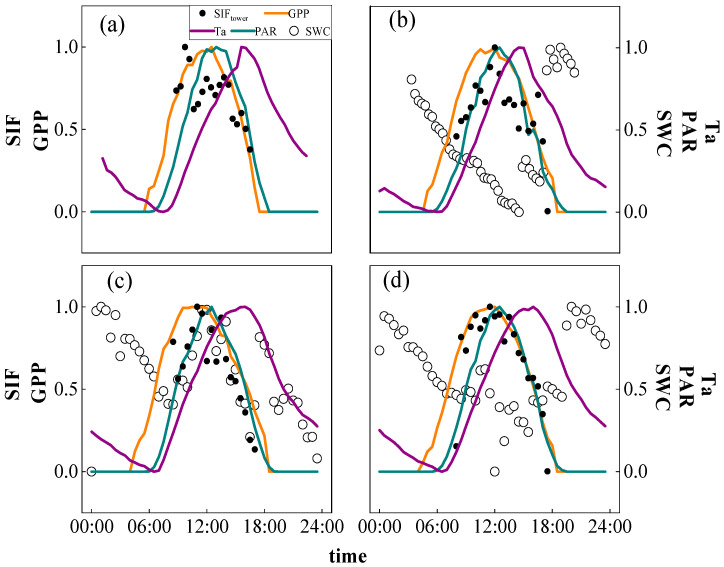
Diurnal variations in SIF, GPP and environmental variables, including gross primary production (GPP, orange line), Solar-induced fluorescence (SIF, black dot), photosynthetically active radiation (PAR, green line), air temperature (Ta, purple line) and soil water content (SWC, open circle), in (**a**) winter; (**b**) spring; (**c**) summer; and (**d**) from January to August in 2021.

**Figure 2 plants-12-02224-f002:**
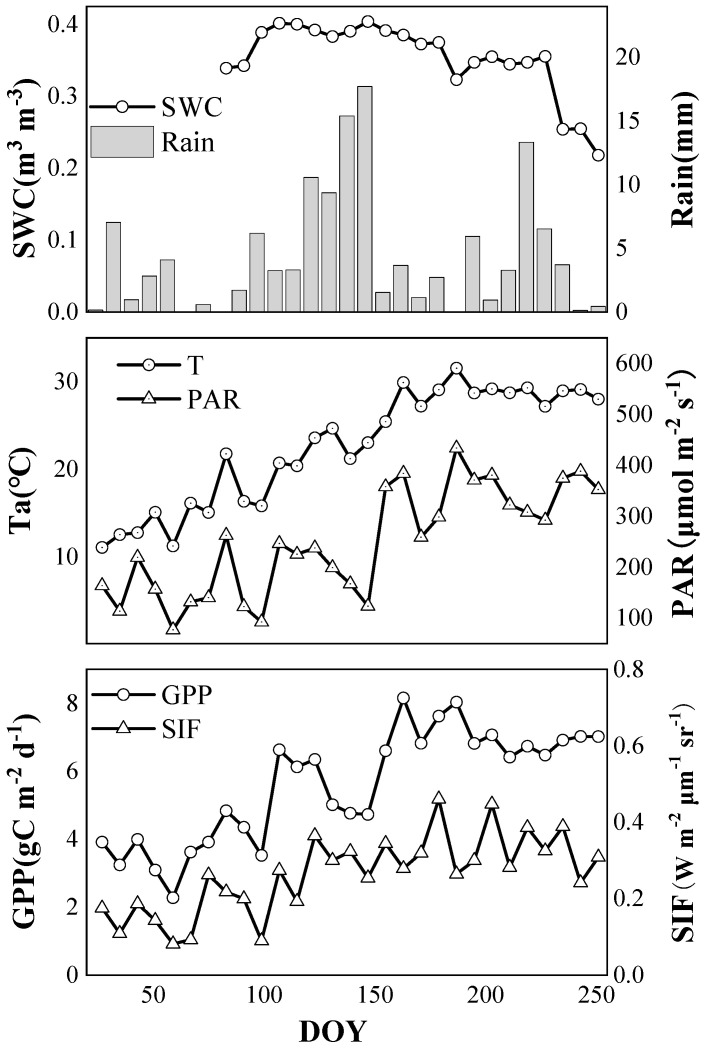
Seasonal variation in SIF, GPP and environmental variables. Rainfall (grey bar) and SWC (open circle) in upper panel; PAR (open triangle) and Ta (open circle) in middle panel; SIF (open triangle) and GPP (open circle) in lower panel.

**Figure 3 plants-12-02224-f003:**
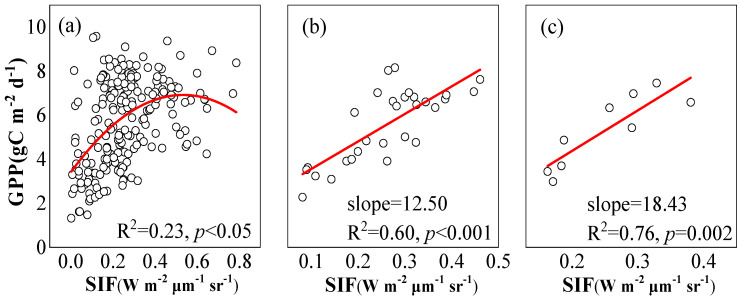
Correlation of SIF-GPP at different time scales: (**a**) daily; (**b**) 8-day; (**c**) monthly.

**Figure 4 plants-12-02224-f004:**
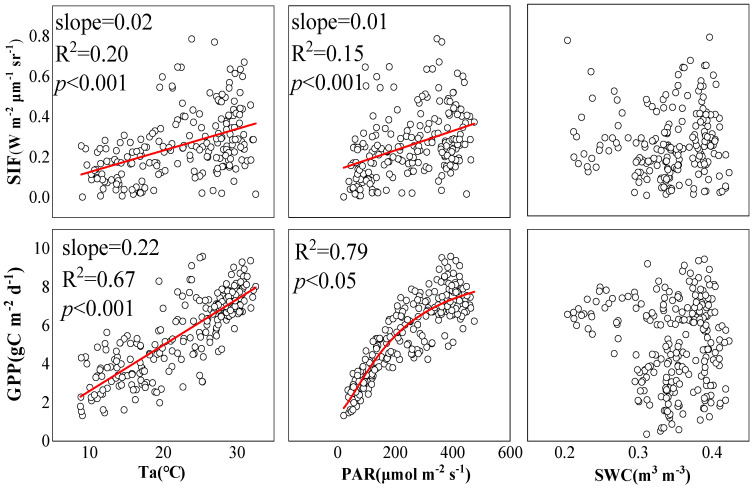
Correlation between SIF, GPP, and environmental variables on daily scale: Each point represents the daily average value.

**Figure 5 plants-12-02224-f005:**
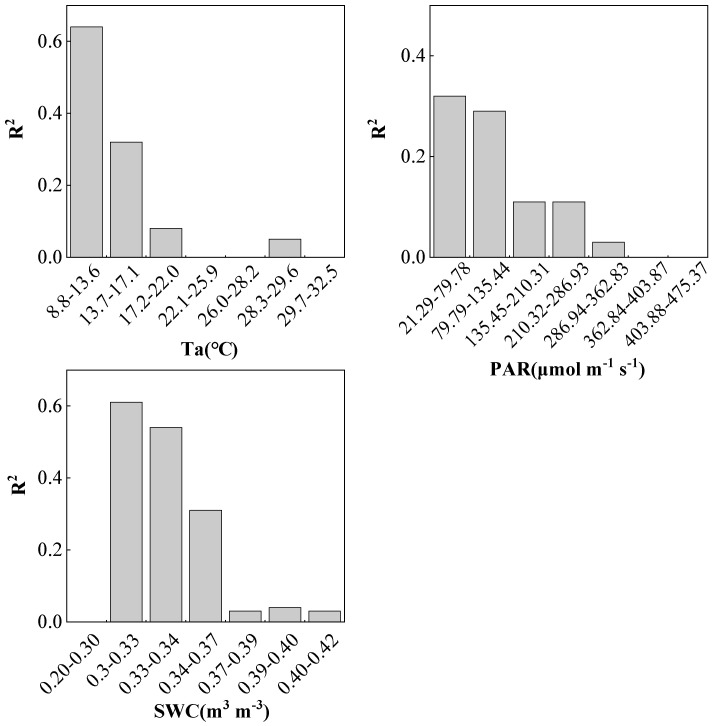
The influence of different environmental variables on SIF–GPP correlation.

**Figure 6 plants-12-02224-f006:**
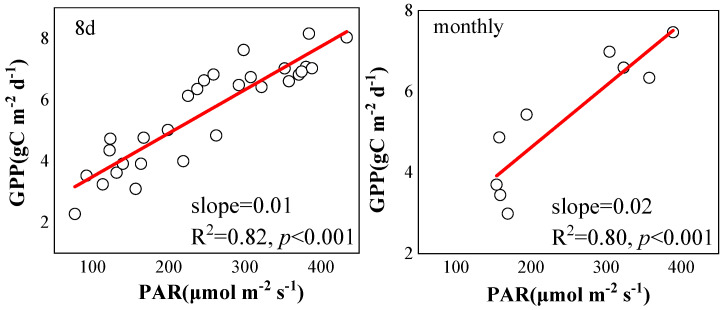
The influence of PAR on GPP on 8-day and monthly scales.

**Figure 7 plants-12-02224-f007:**
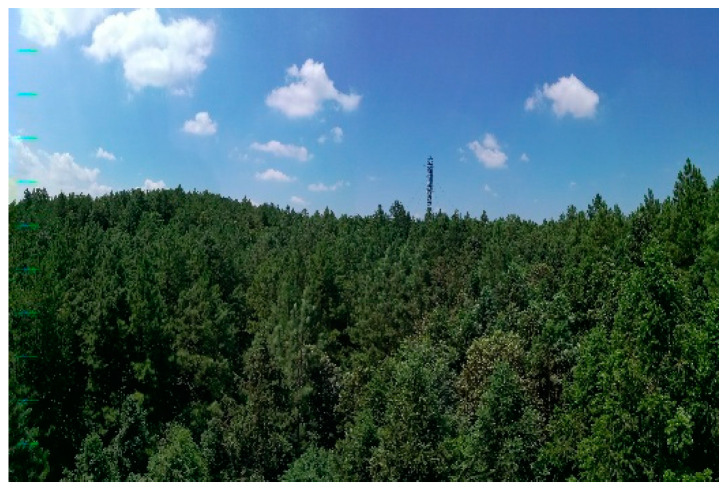
Vegetation landscape of Qianyanzhou Station.

**Figure 8 plants-12-02224-f008:**
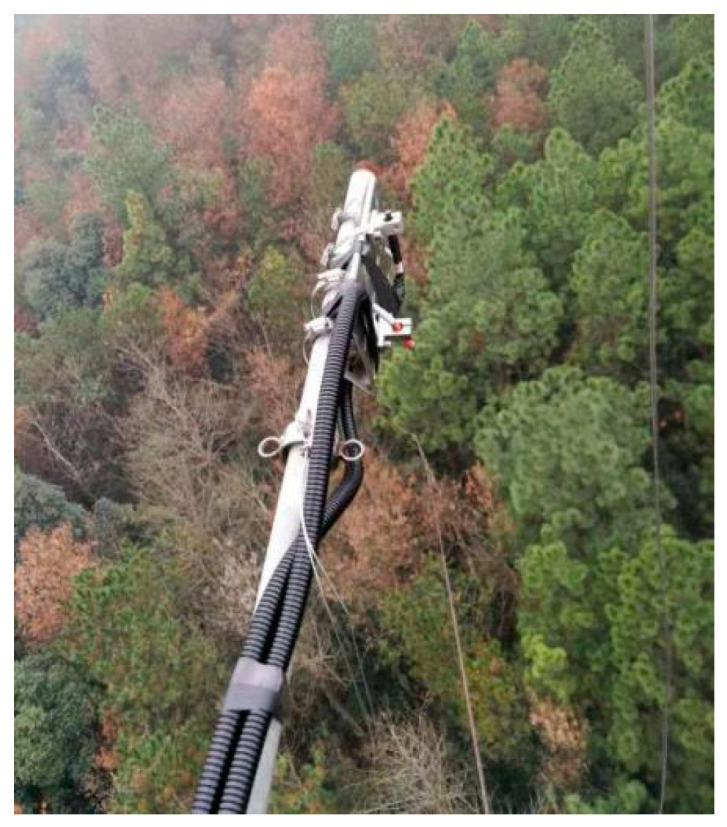
Fluorescent equipment installation.

## Data Availability

Due to the nature of this research, the participants of this study did not agree for their data to be shared publicly; therefore, supporting data are not available.

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
