# Peer review of "Synchronous Changes of GPP and Solar-Induced Chlorophyll Fluorescence in a Subtropical Evergreen Coniferous Forest"

_plants, 2023, doi:10.3390/plants12112224_

Round 1

Reviewer 1 Report

Review comments for Plants-2297101

Title: Synergistic Changes of GPP and Solar-induced Chlorophyll Fluorescence in Subtropical Evergreen Coniferous Forest

Using the measured GPP and SIF of subtropical evergreen coniferous forest in Qianyanzhou station, the authors tried to reveal the correlation between GPP and SIF across different temporal scales and explored their relationships with environmental factors. This study chose an interesting topic and is appropriate for Plants audience. However, this manuscript is not well written and has several significant drawbacks. The reviewer has several concerns and suggestions for the authors to consider:

(1) There are many grammar mistakes in the article, I suggest involving a native speaker to help revise this manuscript.

(2) There are several sentences I don’t understand well. For example,  “air temperature (Ta) and photosynthetically active radiation (PAR) are significant on different time scales.” Please rewrite the abstract.

(3) I suggest having more introduction to SIF, especially the mechanism by which vegetation emits SIF.

(4) Introduce more about the Qianyanzhou Station in the introduction part. In addition, previous GPP and SIF related studies conducted in Qianyan Zhou Station should be introduced.

(5) If the journal “Plant” doesn’t require to place the “Materials and methods” part after the discussion part, I would suggest putting the “Materials and methods” part after introduction, which will help the reader better under this paper.

(6) In Figure 4, we can see both GPP and SIF didn’t have significant relationships with SWC, can you explore the reasons?

(7) Section 3.2: please explain more about the reason for the nonlinear relationship of SIF-GPP, and explore the potential mechanisms why GPP and SIF have a higher R2 when time scale increases.

(8) Section 4.2: please elaborate how did you calculate GPP using NEE.

(9) Please provide some photos of the study place and devices used to measure GPP and SIF.

(10) In Section 5.1, you said “better”, you didn’t have any other indicators of GPP to compare with, how can you say SIF is better?

Author Response

(1)There are many grammar mistakes in the article, I suggest involving a native speaker to help revise this manuscript.

Answer: Thank you for your suggestion. We have invited a native English speaker to help revise this manuscript.

(2)There are several sentences I don’t understand well. For example,  “air temperature (Ta) and photosynthetically active radiation (PAR) are significant on different time scales.” Please rewrite the abstract.

Answer: Thanks for your suggestion. This sentence has been changed to “linear correlation between air temperature (Ta) and photosynthetically active radiation (PAR) with SIF and GPP are significant on different time scales. In our study, the soil water content (SWC) affected the changes in the SIF and GPP of vegetation in Qianyanzhou was not significant”.

(3)I suggest having more introduction to SIF, especially the mechanism by which vegetation emits SIF.

Answer: Thanks for your suggestion.The mechanism by how vegetation emits SIF has been added to the article。

(4)Introduce more about the Qianyanzhou Station in the introduction part. In addition, previous GPP and SIF related studies conducted in Qianyan Zhou Station should be introduced.

Answer: Thanks for your suggestion. The introduction of Qianyanzhou Station has been supplemented and a landscape has been added (Figure 7). In addition, Chen et al. research on SIF at Qianyanzhou Station has been added in Section 1.

(5) If the journal “Plant” doesn’t require to place the “Materials and methods” part after the discussion part, I would suggest putting the “Materials and methods” part after introduction, which will help the reader better under this paper

Answer: Thanks for your suggestion. This is because the journal 'Plant' requires the 'Materials and methods' part after the discussion part, which is why this is done.

(6)In Figure 4, we can see both GPP and SIF didn’t have significant relationships with SWC, can you explore the reasons?

Answer: Thanks for your suggestion. An explanation of the impact of SWC on GPP and SIF has been added in Section 3.

(7)Section 3.2: please explain more about the reason for the nonlinear relationship of SIF-GPP, and explore the potential mechanisms why GPP and SIF have a higher R2 when time scale increases.

Answer: Thanks for your suggestion. The discussion section has been modified and supplemented.

(8) Section 4.2: please elaborate how did you calculate GPP using NEE.

Answer: Thanks for your suggestion. How to calculate GPP using NEE has been added in section 4.2.

(9) Please provide some photos of the study place and devices used to measure GPP and SIF.

Answer: Thanks for your suggestion. The equipment structure and installation are shown in Figure 8.

(10) In Section 5.1, you said “better”, you didn’t have any other indicators of GPP to compare with, how can you say SIF is better?

Answer: Thanks for your suggestion. This sentence has been changed to “The SIF and GPP in Qianyanzhou have similar diurnal and seasonal variation characteristics, and both reach their highest values in summer, indicating that SIF can more effectively represent the seasonal variation in GPP in the ecosystem”.

Reviewer 2 Report

Wang et al. investigated the seasonal changes of GPP and solar-induced chlorophyll fluorescence based on high-frequency near-surface automatic observation system combined with environmental parameters in subtropical evergreen coniferous forest in 2021. They found that Qianyanzhou vegetation SIF and GPP have similar diurnal and seasonal dynamics, and both reach the maximum value in summer, showing that SIF can better track the seasonal dynamics of ecosystem GPP. This study can help to understand the correlation between SIF and GPP in artificial evergreen coniferous forest in the middle subtropical zone. However, the field experiment is unclear. Before being published, some suggestions should be addressed. 1.     The approximate area of the target observation for downward optical fiber? 2.     The ages of the target species should be provided for better understand this field measurement. 3.     How many data points were measured and used for different protocols? 4.     Vegetation physiological status and canopy structure are important for explaining the SIF-GPP relationship. The maximum rate of carboxylation (Vcmax) represents the inherent leaf photosynthetic capacity and varies widely with shifts in season, species, and leaf age. Leaf nitrogen (N) is considered a proxy for Vcmax because N is the main component of photosynthetic enzymes and directly participates in the photosynthesis process as a catalyst. Nitrogen allocation regulates the relationship between maximum carboxylation rate and chlorophyll content along the vertical gradient of subtropical forest canopy. 5.     This is an interesting finding that SIF reach the maximum value by the end of June while GPP reach the maximum value at the beginning of June. Add more literature to discuss this difference. 6.     Conclusions were not the repeat of the results, I suggest rewrite this section.  

Author Response

1.The approximate area of the targetobservation for downward optical fiber?

Answer: Thank you for your suggestion. The approximate area of the ground measured by the downward optical fiber of the equipment is about 100m2.

2.The ages of the target species should be provided for better understand this field measurement.

Answer: Thank you for your suggestion. The target species has a forest age of approximately 40 years.

3.How many data points were measured and used for different protocols?

Answer: Thank you for your suggestion. Due to equipment limitations, this study only selected one experimental site at Qianyanzhou Station for observation of chlorophyll fluorescence and flux, which is the sunny part. When installing measuring instruments, it is recommended to keep the ground vegetation unobstructed as much as possible. Afterwards, more experimental points and measurement data from other sites will be added for more in-depth comparative research.

4.Vegetation physiological status and canopy structure are important for explaining the SIF-GPP relationship. The maximum rate of carboxylation (Vcmax) represents the inherent leaf photosynthetic capacity and varies widely with shifts in season, species, and leaf age. Leaf nitrogen (N) is considered a proxy for Vcmax because N is the main component of photosynthetic enzymes and directly participates in the photosynthesis process as a catalyst. Nitrogen allocation regulates the relationship between maximum carboxylation rate and chlorophyll content along the vertical gradient of subtropical forest canopy.

Answer: Thanks for your suggestion. The discussion section has been modified and supplemented.

5.This is an interesting finding that SIF reach the maximum value by the end of June while GPP reach the maximum value at the beginning of June. Add more literature to discuss this difference.

Answer: Thanks for your suggestion. This study mainly explores whether there is consistency between SIF and GPP in daily and seasonal changes. After research and analysis, the SIF and GPP of vegetation at Qianyanzhou Station have good consistency in daily and seasonal changes, but there are still differences in their changes. One of the reasons for this situation may be due to the different ways of obtaining SIF and GPP, as well as the errors that occur during data processing. The difference in seasonal variation between SIF and GPP is indeed an interesting finding, and we will delve deeper into the underlying reasons in future research.

6.Conclusions were not the repeat of the results, I suggest rewrite this section. 

Answer: Thanks for your suggestion. The discussion section has been modified and supplemented.

Reviewer 3 Report

Wang and Zhang examined the SIF-GPP relationship in a subtropical needle leaf forest at different time scales. It is interesting that the research incorporates the data collected from a tower-based spectrometer. However, the manuscript tries to discuss too many determining factors for SIF-GPP. I find it difficult to follow the focus of this research. I think this manuscript needs to be revised before being considered for publish. 

Major comments:

  1. Any reason to put the methods at the end? It is hard to understand the results without knowing how the data are collected.

  2. The data processing section needs more specific. I cannot get what the data represent for in all figures from this short paragraph on data processing. 

  3. For the spectrometer, what is the field of view? How much area does the spectrometer observe?

  4. What is the difference between sections 3.2 and 3.3? The two subtitles seems to tell similar content.

Minor comments:

Line 18: “significant” for what?

Line 19: Does the “soil” effect means soil reflectance or soil moisture? I think you mean the latter, please specify. 

Line139 and all figure, please be more explanatory in figure captions. For example, are the daily variations in figure 1 from seasonal averages? In figure 4, does each point represent an annual mean at daily scale (7-minute interval)?

Line 254:Vcmax doesn’t change within a day. Please clarify this sentence.

Line 321: solar incidence appears twice here. What is the correct third signal?

Author Response

Major comments:

1.Any reason to put the methods at the end? It is hard to understand the results without knowing how the data are collected.

Answer: Thanks for your suggestion. This is because the journal 'Plant' requires the 'Materials and methods' part after the discussion part, which is why this is done.

2.The data processing section needs more specific. I cannot get what the data represent for in all figures from this short paragraph on data processing.

Answer: Thanks for your suggestion. This section has been modified.

3.For the spectrometer, what is the field of view? How much area does the spectrometer observe?

Answer: Thanks for your suggestion. The approximate area of the ground measured by the downward optical fiber of the equipment is about 100m2.

4.What is the difference between sections 3.2 and 3.3? The two subtitles seems to tell similar content.

Answer: Thanks for your suggestion. Section 3.2 attempts to explain the underlying mechanism of a more linear correlation between GPP and SIF as time scales increase.  Section 3.3 attempts to explain the impact of different environmental factors on the correlation between SIF and GPP. This study has revised Section 3 to make it more comprehensive.

Minor comments:

Line 18: “significant” for what?

Answer: Thanks for your suggestion. Linear correlation between air temperature (Ta) and photosynthetically active radiation (PAR) with SIF and GPP are significant on different time scales.

Line 19: Does the “soil” effect means soil reflectance or soil moisture? I think you mean the latter, please specify. 

Answer: Thanks for your suggestion. The “soil” effect means soil moisture.

Line139 and all figure, please be more explanatory in figure captions. For example, are the daily variations in figure 1 from seasonal averages? In figure 4, does each point represent an annual mean at daily scale (7-minute interval)?

Answer: Thanks for your suggestion. It has been explained in figure captions.

Line 254:Vcmax doesn’t change within a day. Please clarify this sentence.

Answer: Thanks for your suggestion. The discussion section has been modified and supplemented.

Line 321: solar incidence appears twice here. What is the correct third signal?

Answer: Thanks for your suggestion. The instrument adopts a "sandwich" mode measurement method, where the spectrometer records two downward solar input spectral signals and one upward ground reflection spectral signal in the order of solar incidence, ground reflection, and solar incidence.When recording the second solar incident spectrum signal, a set of "sandwich" data recording is completed. After a 7-minute interval, proceed to the next set of data measurements.

Round 2

Reviewer 1 Report

Review comments for Plants-2297101-v2

Title: Synergistic Changes of GPP and Solar-induced Chlorophyll Fluorescence in Subtropical Evergreen Coniferous Forest

The authors have revised the manuscript according to the reviewer’s suggestions. However, the quality of this paper still needs to be improved before it can be published.

First, the abstract is not well written. There are several grammatical errors in the abstract. For example: line 20-21: “In our study, the soil water content (SWC) affected the changes in the SIF and GPP of vegetation in Qianyanzhou was not significant.” Th abstract needs to be rewritten.

Second, you didn’t provide convincing explanations as to why GPP and SIF didn’t have significant relationships with SWC.

Third, please elaborate how did you use NEE to calculate GPP. Please elaborate your flux data processing and quality control. It is important to this paper. Otherwise, I don’t think you really know how your GPP data were calculated.

Line 39: the reference is missing.

Author Response

First, the abstract is not well written. There are several grammatical errors in the abstract. For example: line 20-21: “In our study, the soil water content (SWC) affected the changes in the SIF and GPP of vegetation in Qianyanzhou was not significant.” Th abstract needs to be rewritten.

Answer: Thank you for your suggestion. The abstract has been modified, and this sentence has been changed to 'the linear correlation between soil water content (SWC) with SIF and GPP are not significant'.

Second, you didn’t provide convincing explanations as to why GPP and SIF didn’t have significant relationships with SWC.

Answer: Thank you for your suggestion. The reasons for the insignificant impact of SWC on SIF and GPP have been improved and modified in the article. SWC is an important indicator that reflects whether vegetation is under drought stress. In this study, due to the sufficient soil moisture content at Qianyanzhou Station, vegetation was not significantly affected by drought stress, and the photosynthetic rate and chlorophyll fluorescence production of vegetation were not significantly affected by drought. In addition, because the station is in a subtropical monsoon climate with abundant precipitation, this study shows that SWC is greatly affected by Rain, and its daily and seasonal changes are not obvious. Therefore, SWC has no significant correlation with vegetation GPP and SIF

Third, please elaborate how did you use NEE to calculate GPP. Please elaborate your flux data processing and quality control. It is important to this paper. Otherwise, I don’t think you really know how your GPP data were calculated.

Answer: Thank you for your suggestion. This study directly utilized the shared data of ChinaFLUX. Currently, research work both domestically and internationally follows similar data quality control methods and processing processes, including abnormal data screening, WPL correction, coordinate axis rotation, friction wind speed screening, and missing data interpolation. According to expert opinions, this section has been supplemented and improved by referring to literature such as Dai et al.(2021).

Line 39: the reference is missing.

Answer: Thank you for your suggestion. Reference has been added.

Reviewer 3 Report

I think the revision is well done and ready to publish. Thanks for incorporating my comments in the manuscript. Please make sure all references are included, e.g., line 39.

Author Response

I think the revision is well done and ready to publish. Thanks for incorporating my comments in the manuscript. Please make sure all references are included, e.g., line 39.

Answer: Thank you for your suggestion. Reference has been added.

Round 3

Reviewer 1 Report

Review comments for Plants-2297101-v3

Title: Synergistic Changes of GPP and Solar-induced Chlorophyll Fluorescence in Subtropical Evergreen Coniferous Forest

The authors have revised the manuscript according to the reviewer’s suggestions. And the quality of manuscript has been improved. The reviewer still has one suggestion:

Please add several related references to support your explanations as to why GPP and SIF didn’t have significant relationships with SWC.

Author Response

Please add several related references to support your explanations as to why GPP and SIF didn’t have significant relationships with SWC.

Answer: Thanks for your suggestion. The reasons for the insignificant impact of SWC on SIF and GPP have been improved and modified in the article. In addition to the reasons mentioned earlier in the paper, research has shown that SIF mainly responds to changes in PAR, rather than actual changes in photosynthetic efficiency. The key driving factor for SIF and GPP is photosynthetic effective radiation. SWC is an important indicator that reflects which vegetation is under drought stress. Previous research shows that under drought stress, the plant photosynthesis efficiency decreases due to the reduction of transpiration rate of leaves, so the impact of SWC on SIF and GPP is not significant. However, there are still many deficiencies in this study, such as how SIF characterizes the changes in GPP of vegetation at Qianyanzhou Station under extreme drought and high temperature stress. Further research will be done based on longer data analysis in the future.